# A Smooth Transition Algorithm for Adjacent Panoramic Viewpoints Using Matched Delaunay Triangular Patches

**Pengcheng Zhao**, **Qingwu Hu**, **Zhixiong Tang †** and **Mingyao Ai** *

School of Remote Sensing and Information Engineering, Wuhan University, Wuhan 430072, China;
pengcheng.zhao@whu.edu.cn (P.Z.); huqw@whu.edu.cn (Q.H.); dvorak4tzx@gmail.com (Z.T.)
* Correspondence: aimingyao@whu.edu.cn
† Current address: Beijing Momenta Technology Co., Ltd., Beijing 100089, China.

**Abstract:** The unnatural panoramic image transition between two adjacent viewpoints reduces the immersion and interactive experiences of 360° panoramic walkthrough systems. In this paper, a dynamic panoramic image rendering and smooth transition algorithm for adjacent viewpoints is proposed. First, the feature points of adjacent view images are extracted, a robust matching algorithm is used to establish adjacent point pairs, and the matching triangles are formed by using the homonymous points. Then, a dynamic transition model is formed by the simultaneous linear transitions of shape and texture for each control triangle. Finally, the smooth transition between adjacent viewpoints is implemented by overlaying the dynamic transition model with the 360° panoramic walkthrough scene. Experimental results show that this method has obvious advantages in visual representation with distinct visual movement. It can realize the smooth transition between two indoor panoramic stations with arbitrary station spacing, and its execution efficiency is up to 50 frames per second. It effectively enhances the interactivity and immersion of 360° panoramic walkthrough systems.

**Keywords:** panoramic image; scene transition; mesh morph; image-based rendering; texture shade

## 1. Introduction

A 360° panoramic walkthrough system is a kind of virtual reality technology based on the sequenced panoramic images. The image-based method enables walking through space much more easily than using 3D modeling and rendering. It is widely used in 3D visualization, virtual reality and augmented reality because of its easy data acquisition and processing, low bandwidth requirements, high resolution, strong authenticity and good navigation on cheap hardware display devices [1]. Panoramic view is a new type of map service that displays 360° panoramic images taken in cities, streets, museum and other places in the form of a 3D street view based on 360° panorama technology. Users can use street view services to obtain an immersive experience by roaming around the 360° and 3D simulation scene without visiting the place. These services overcome the shortcomings of a traditional map, which is incapable of delivering a real scene. Thus, street view services are highly significant for applications research [2].

Since panoramic image acquisition occurs on some limited and discontinuous sites, there are two main problems in a 360° panoramic walkthrough system compared to a general virtual reality system. First, the observation is limited to a few specific roaming points. Users can not freely move in the scene, and wandering can be realized only when there is only one viewpoint in the scene. Second, skipping occurs between roaming points, and the transition is unnatural, which seriously affects the

walkthrough experience [3,4]. It is difficult to solve the previous problem without increasing the density of panorama stations and building a true 3D model. The latter problem can be compensated for by the visual effect of the smooth transition between the viewpoints, which is also the subject of this paper.

The smooth transition between the viewpoints can be used in compressing video sequences with high compression rates, virtual tourism and entertainment, 3D scene displays of real estate and public security, etc. The smooth transition algorithm for adjacent panoramic viewpoints is crucial for improving the user experience of the 360° panoramic roaming system, the practicability of indoor panoramic exhibition applications, and the usability of indoor panoramic navigation applications.

By summarizing the most current panoramic systems, there are several kinds of visual smoothing methods used in panoramic site transition:

(1) Skipping directly from one panorama site to another dismembers the relationships between the panoramic sites. The large flicker in the scenes can disorient the user. Unfortunately, most of the existing panoramic walkthrough systems use this skip mode to transition between adjacent sites without any strategies to mitigate this poor visual experience [5]. To achieve a smooth transition between adjacent panorama sites, four common transition methods have been studied in previous research, including the texture transparent gradient, noise interference, model stretching and parallax.

(2) The texture-transparent gradient method reduces the flickering by linearly interpolating a series of panoramic images and fusing the colors of the two panorama sites [6]. This method is simple and easy to implement, and it has steady visual effects at the beginning and end of the transition. However, an obvious ghosting occurs when the transition approaches half. In some game scenarios, noise interference is a common method of scene transition. The method disperses the user's attention by generating some "noise" in the field of vision; this distracts the user's attention from the unnatural scene transition, and the "noise" is usually the jitter, distortion, or change in the hue of the pixels on the screen [7]. The method is essentially a visual deception, and the effect is limited.

(3) The model stretching method is based on a simple assumption that a feeling of "going in" is available by stretching the scene from the center of the view to the periphery, since the display effect will be magnified to different degrees as the distance from the objects in the visible area to the viewpoint is closer during the transition between sites. It can give the user a sense of moving forward using a simple image stretching process. This method can be applied to outdoor scenes with a broad perspective and relatively wide distances from objects to the viewpoint, especially in the street view with a layout arranged along the road [8,9]. However, because of the greater space shielding and smaller distance from objects to the viewpoint indoors, the stretching treatment will give an exaggerated sense of space displacement that is unnatural and unrealistic.

(4) In recent years, parallax based on human eye perception has been introduced into panoramic image roaming to improve the transitional effect, which includes full parallax, tour into the picture and fake parallax [10,11]. The full parallax method requires the support of fine 3D models, but the general panoramic walkthrough system does not have the necessary data. The fake parallax method only extracts the rough geometric information and the color layer of the background, and it simulates the parallax effect using a simple translation and zooming. Parallax is used mainly to smooth the observations from moving around while keeping eyes on an object. There are some marked differences in the direction and scale of motion between moving around and panorama scene transition.

To solve the problems of the destroyed spatial relationships among the panorama stations, and the disorientation caused by the inability of the user to perceive the change in position during panorama scene transition, a dynamic panoramic image rendering algorithm for smooth transitions between adjacent viewpoints in indoor scenes is proposed in this paper. Based on the principle of using

homonymous points to guide graphics deformation, pairs of matched triangular patches are formed by homonymous points; then, a smooth interpolation of shape and texture is made with triangular patches as units to dynamically generate a panorama for each frame. This method dynamically constructs the sequence panorama with motion sensing during the transition. This makes the scene transition as real as walking in a three-dimensional scene, and effectively avoids the spatial disorientation.

Experiments using indoor panoramic images collected by different equipment and different distances between stations are performed. The results show that the proposed method can realize adaptive smooth transition between different scenes with a rendering efficiency of nearly 50 fps (frames per second), which effectively enhances the interactivity and immersion of the panoramic roaming system. Compared with similar research work, the special points of this paper are as follows:

(1) In the indoor space with a relatively regular layout but serious occlusion, instead of drawing matching points manually [12], we construct matched triangular patches by using the method of obtaining feature points and lines after panoramic orthophoto projection.

(2) We use barycentric coordinates to perform the panoramic transition algorithm directly on the spherical panoramic model, instead of the transition of the cylindrical panoramic image [13] and the cube panoramic image [14].

(3) The idea of our panoramic transition algorithm is similar to [15], but we constructed a panoramic system with a panoramic deformation layer, and analyzed its visual effect, practicability and operation performance in detail.

To illustrate the problem about the unnatural panoramic transition and better understand the dynamic panorama transition method, the videos named "problem of panoramic transition.mp4" (https://youtu.be/R_lW1Xz8QNc) and "smooth panoramic transition using matched triangular patches.mp4" (https://youtu.be/xQyqiUuLPCI) are provided as an attachment for readers.

## 2. Methods

To overcome the jumping between the panoramic images of adjacent viewpoints, the graphics deformation principle is presented for the matched triangle in the adjacent panoramic images. The panoramic images are divided into many matched triangular patches. The matched triangular patches are then taken to implement graphical deformation to realize smooth transitions between the panoramic images of adjacent viewpoints. The schematic diagram of the basic principle is shown in Figure 1. Transitional frames are generated by the smooth interpolation of the shapes and textures of all matched triangular patches.

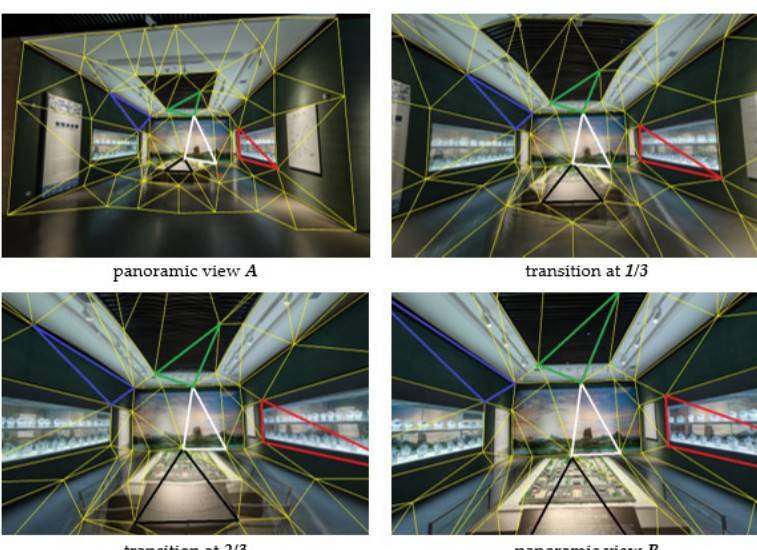

**Figure 1.** Principle schematic diagram of panorama transition guided by matched triangular patches.

As Figure 1 shows, the panorama scene transitions from the current site to the next site are controlled by the transition of yellow triangular patches with a blended texture that are formed by fading out the current panorama A and fading in the next panorama B. The white triangle in the center of the view is enlarged with a slight deformation, but the red, blue, green, and black triangles are obviously stretched because of the transverse Mercator projection.

To achieve the smooth panorama transition between adjacent viewpoints with frame rates of no less than human visual perception, the production workflow is designed as shown in Figure 2. The proposed method is divided into three parts. First, the triangular patches for controlling transitions are generated from the feature points of panoramas by a triangulation algorithm featured by maximizing the minimum angles. The above work is detailed in Section 2.1. Then, we morph the triangular patches on the surface of a 3D panorama sphere while fading out texture A and fading in texture B. The principles and algorithms involved in this section are presented in Sections 2.2 and 2.3. Finally, a strategy for implementing the panoramic walkthrough system, which is equipped with a dynamic panoramic image rendering algorithm for smooth transitions between adjacent viewpoints by overlying the morph model layer, is described in Section 2.4.

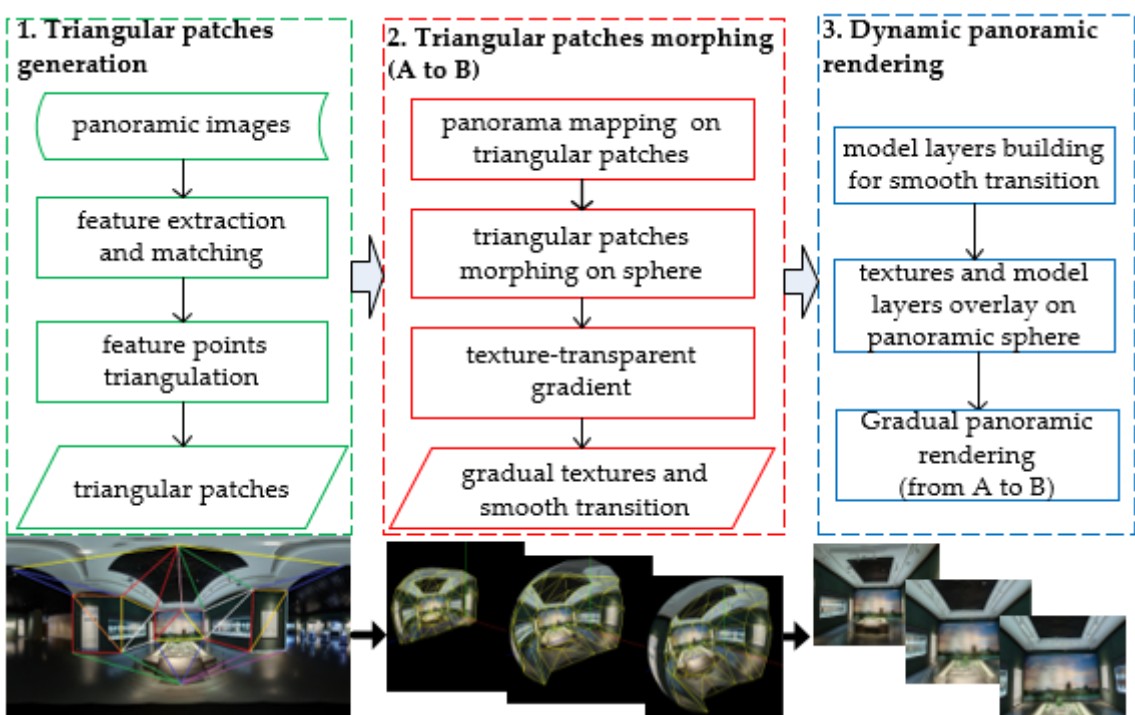

**Figure 2.** Proposed technique workflow for smooth transition between adjacent viewpoints.

## 2.1. Generation of Matched Delaunay Triangular Patches

The solutions of the panoramic image matching problem can be roughly divided into three categories. First, Bay et al. [16] directly applied the existing techniques for general frame image processing to panoramic images. The algorithm is simple, but the special geometric deformation of the panoramic image has a profound influence on the matching results. Second, the feature extraction and various operations are directly performed in the spherical scale space, and the matching accuracy is high, but the model is complex and difficult to implement. The rigorous spherical geometry model was adopted to map the panoramic image onto the sphere. The image function was expressed in the form of a spherical harmonic function, and various operations are carried out in the spherical scale space in [17]. Third, Mauthner et al. [18] proposed a virtual imaging surface for matching that is generated by resampling the panoramic image according to the perspective projection model, but the obtained feature matching is relatively limited.

In this paper, the way to obtain homonymous points referenced in the method of the virtual imaging surface (the third solution above) is shown in Figure 3.

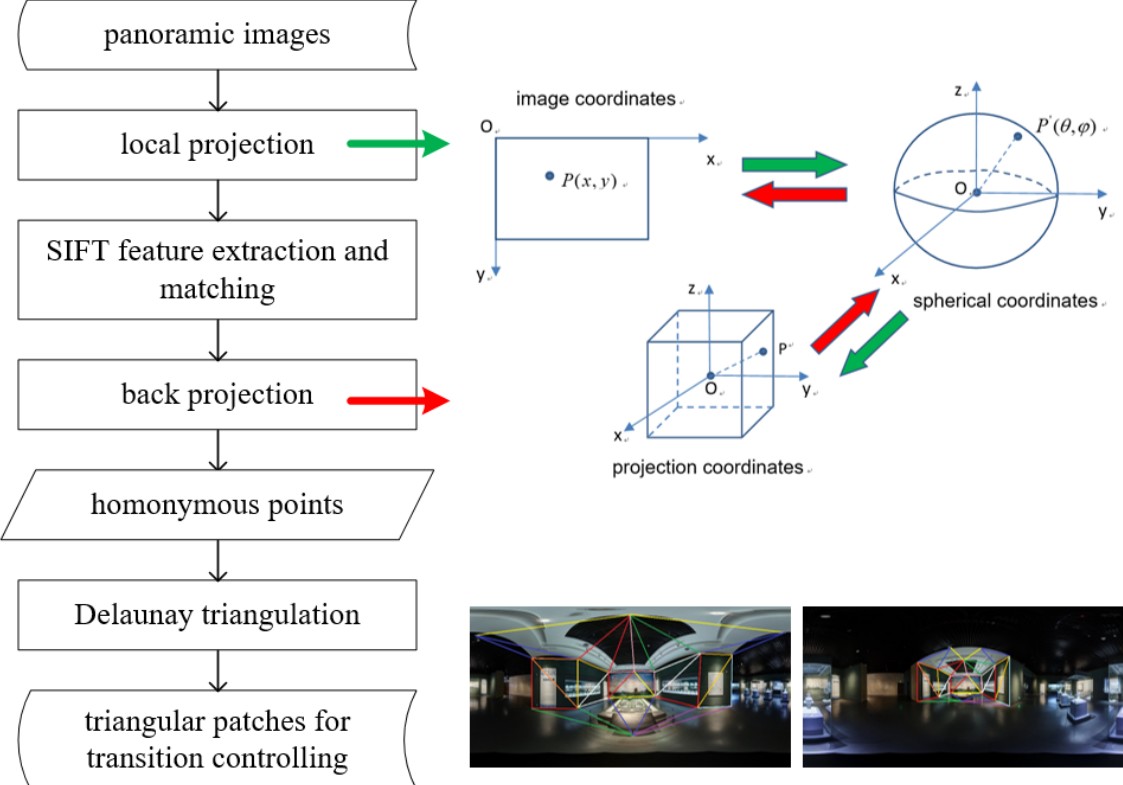

**Figure 3.** Generation flow chart of triangular patches for controlling transition. The green arrow represents the direction of execution of the local projection, and the red arrow represents the direction of execution of the back projection. The picture at the bottom right shows the extracted control triangular patches.

First, the panoramic image is projected onto a cube to form six local planes, and the homonymous feature points are extracted by the classical SIFT (Scale-invariant feature transform) on all local planes. Then, a following back projection converts these points back into the image coordinates. Finally, triangular patches are generated by the triangulation algorithm from homonymous points. Based on the preferred homonyms points, the Delaunay triangulations algorithm is used to generate the triangular patches [19]. It tends to avoid sliver triangles because it maximizes the minimum angle of all the angles of the triangles in the triangulation. Moreover, a Delaunay triangulation for a given set P of discrete points in a plane is a triangulation DT(P) such that no point in P is inside the circumcircle of any triangle in DT(P). This ensures that no control faces intersect.

## 2.2. Synchronous Morphing of the Shape and Texture

Based on the idea of the transition guided by control points, we use triangular patches as the basic control unit to transform the adjacent panorama. As the simplest surface element, a triangle's transition is shown in Figure 4. With the increase from 0 to 1, a triangle's deformation is accomplished by simultaneously changing the triangle shape and the interior point texture. The original triangle (red) is displayed on the left, the target triangle (blue and green) is displayed on the right, and the gradual triangles (from 1/4 to 1/2 to 3/4) become narrower and greener. The solid black spots indicate the movement of the triangle's vertices, and the hollow spots express the textural interpolation of interior points.

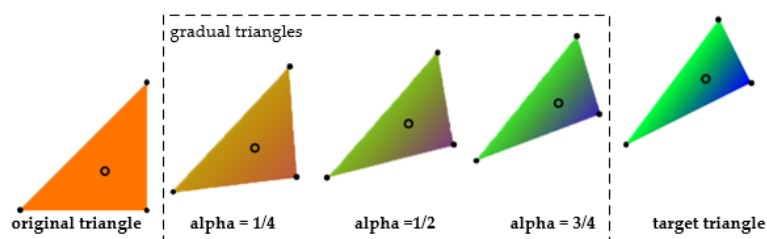

**Figure 4.** Demonstration of the transition from one tri angle to another.

When a triangular surface is deformed, the positions and textures of its interior points need to be changed. In the Cartesian coordinate system, this operation requires a large number of interpolated calculations. However, in the Barycentric coordinates system, this operation becomes easy due to an important property in which the barycentric coordinates of each point in the triangle remain during a linear deformation [20].

According to the definition of the Barycentric coordinates, considering a triangle $\Delta ABC$ defined by its three vertices, $A(x_1, y_1)$, $B(x_2, y_2)$ and $C(x_3, y_3)$, any interior point $p$ in $\Delta ABC$ can be expressed as a linear combination of three vertex coordinates $p = \lambda_1 * A + \lambda_2 * B + \lambda_3 * C$, where $\lambda_1, \lambda_2, \lambda_3 \geq 0$ and $\lambda_1 + \lambda_2 + \lambda_3 = 1$; the combination of parameters $(\lambda_1, \lambda_2, \lambda_3)$ is regarded as the Barycentric coordinates of point $p$. Specially, the vertices themselves have the coordinates $A = (1, 0, 0)$, $B = (0, 1, 0)$ and $C = (0, 0, 1)$.

Setting an interior point $p$ with the Cartesian coordinates $(x, y)$ and the Barycentric coordinates $(\lambda_1, \lambda_2, \lambda_3)$, there is the following deduction:

(1)  The transformation equation from Barycentric coordinates to Cartesian coordinates can be described as follows:

$$\left[\begin{array}{c} x \\ y \end{array}\right] = \left[\begin{array}{c} \lambda_1 x_1 + \lambda_2 x_2 + \lambda_3 x_3 \\ \lambda_1 y_1 + \lambda_2 y_2 + \lambda_3 y_3 \end{array}\right], \tag{1}$$

(2)  The transformation equation from Cartesian coordinates to Barycentric coordinates can be described as follows:

$$\left[\begin{array}{c} \lambda_1 \\ \lambda_2 \end{array}\right] = \left[\begin{array}{c} \frac{(y_2 - y_3)(x - x_3) + (x_3 - x_2)(y - y_3)}{((y_2 - y_3)(x_1 - x_3) + (x_3 - x_2)(y_1 - y_3)} \\ \frac{(y_3 - y_1)(x - x_3) + (x_1 - x_3)(y - y_3)}{(y_2 - y_3)(x_1 - x_3) + (x_3 - x_2)(y_1 - y_3)} \end{array}\right] \tag{2}$$

The synchronous shape morph and the texture interpolation of a triangle can be realized by Algorithm 1 with the following steps.

1.  At the moment that $a$ in triangle $\Delta ABC$ transitions to a new position in $\Delta A'B'C$, the coordinates of three vertices are the linear interpolation of the beginning and end positions in the Cartesian coordinates system as in Equation (3).

$$P_\alpha = P_o + \alpha * (P_t - P_o), \tag{3}$$

where $P_o$ is the beginning coordinate, $P_t$ is end coordinate, and $P_\alpha$ is the coordinate at the moment $\alpha$.

2.  For each interior point $v$ in $\Delta ABC$, a blended texture should be calculated. To blend the texture of the origin triangle $T_o$ with the texture of target triangle $T_t$, the corresponding position of $v$ in $\Delta A'B'C'$ should be calculated by Equations (1) and (2). After we get obtain the Barycentric coordinate $v_b$ of $v_o$ in $\Delta ABC$, the Cartesian coordinate $v_t$ of $v_b$ in $\Delta A'B'C'$ can then be obtained.

3.  The texture of the interior point $v$ can be interpolated according to the corresponding texture $T_o$ and $T_t$ of $v_o$ and $v_t$. The color interpolation follows Equation (4).

$$T_\alpha = (1 - \alpha) * T_o + \alpha * T_t, \tag{4}$$

4.  A temporary triangle is meshed by the new shape and blended texture at moment $\alpha$.

---

**Algorithm 1.** Synchronously Morph the Shape and Interpolate the Texture of a Triangle

---

**Input:** the origin triangle $P_o$, and the target triangle $P_t$, the progress $\alpha$
**Begin**
　　interpolate three vertices of triangle $P_a$ using Equation (3)
　　**for** each interior point $V$ in $P_o$ **do**
　　　$V_{before}$ is the Cartesian coordinate of $V$ before the transition
　　　$V_{middle} = Cartesian2Barycentric\left(V_{before}\right)$, in $P_o$
　　　$V_{after} = Barycentric2Cartesian(V_{middle})$, in $P_a$
　　　$T_{new} = Equation4(\alpha, V)$, the texture at $V$
　　**end**
　　build new mesh $M_a$ by the $V_{after}[]$ and $T_a[]$
**End**

---

### 2.3. Transition between Adjacent Panoramic Viewpoints

The transition from panoramic site A to panoramic site B can be accomplished by dividing panoramic scenes into corresponding patches (Section 2.1), and then performing morphing operations (Section 2.2) on these patches. The process of transition from panoramic site A to B follows Algorithm 2.

---

**Algorithm 2:** Transition the Panorama from A to B

---

**Input:** two site panorama images $I_A$, $I_B$, the control triangular patches $T$
**Begin**
　　mapping $I_A$ and $I_B$ to $T_j$, $j \in \{1..M\}$
　　initial state $Current\_View = I_A$
　　**Loop**: progress $a$ from 0.0 to 1.0
　　　**for** each triangle $P$ in $T$ **do**
　　　　$F_a += Algorithm1(P)$ on Spherical coordinate
　　　**end for**
　　　$Current\_View = F_a$.
　　**end loop**
　　final state $Current\_View = I_B$
**End**

---

1.  Initially, panoramas of the current station $I^{(a)}$ and the next station $I^{(b)}$ are mapped to each $T_j$ to initialize the triangular patches. The current view shows the panoramic site A, meaning that $I^{(a)}$ is fully visible and $I^{(b)}$ is completely transparent.
2.  At each moment $\alpha$ during the transition (more than 50 frames), each triangle in the control triangular patches transitions to a new status using Algorithm 1. Stitching all new status triangle together, a current frame $F_a$ was formed to update the panoramic view.
3.  The end-state shows the panoramic site B with fully visible $I^{(b)}$ and completely transparent $I^{(a)}$.

The middle textures $I^{(a)\rightsquigarrow(b)}$ are formed by collecting all $F_a$. Figure 5 shows a sample of the middle textures generated by the smooth transition algorithm using the matched Delaunay triangular patches of two adjacent viewpoints.

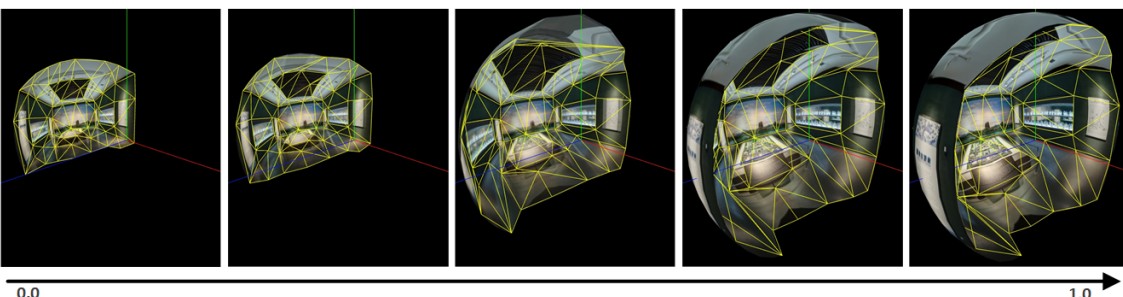

**Figure 5.** Middle textures generated by the smooth transition algorithm. The panorama scene transitions from the current site to the next with the process ranging from 0.0 to 1.0. The control patches transition from a small piece on the sphere to a hemi-spherical surface as the viewpoint moves forward.

The extra fact to note is that a spherical interpolation function is used instead of a linear interpolation function to morph the triangles in the spherical space. According to Liem [21], the linear interpolation on longitude and latitude coordinates is equivalent to the spherical interpolation on Cartesian coordinates when the pixel coordinates are proportional to the spherical longitude and latitude on the panorama with an aspect ratio of 2:1. Therefore, the morphing mesh's transition can fully perform based on Section 2.2.

### 2.4. Dynamic Panoramic Image Rendering

According to the transitional method described in Section 2.3, a set of dynamic textures $I^{(a)\rightsquigarrow(b)}$ was collected. Therefore, the natural transitional effect of the panoramic scene can be achieved by "playing" the dynamic textures $I^{(a)\rightsquigarrow(b)}$ on the panoramic ball. This strategy is easy to implement but is inefficient, because it requires the constant formation of new textures and retexture mapping, which on a large image requires serious CPU and memory consumption.

The dynamic panoramic ball display strategy was proposed as shown in Figure 6. To avoid repeated texture mapping, a morphing mesh is added between the experiencer's eyes and the regular panoramic ball as a model layer. Therefore, the panorama of site A or B is displayed to the users when roaming a fixed site, and the morphing mesh is displayed to users when jumping between sites.

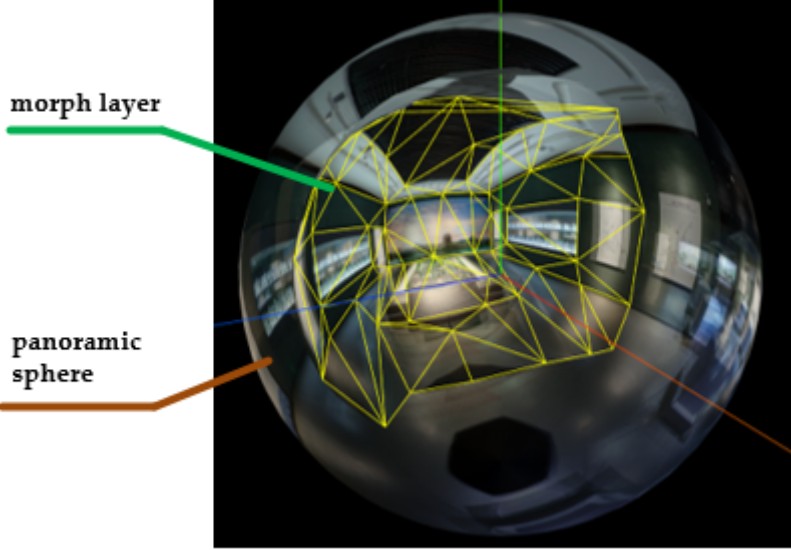

**Figure 6.** The dynamic panoramic ball display strategy. The outer layer is a panoramic sphere for roaming, and the inner layer is the morph layer for smooth site transitions.

The implementation of the dynamic panoramic image rendering algorithm for smooth transitions between adjacent viewpoints based on matched Delaunay triangular patches is shown in Figure 7. More specifically, a spherical triangular mesh was generated by three-dimensional Delaunay triangulations. Then, the dynamic morph layer was implemented by transitioning between adjacent panorama sites on the sphere.

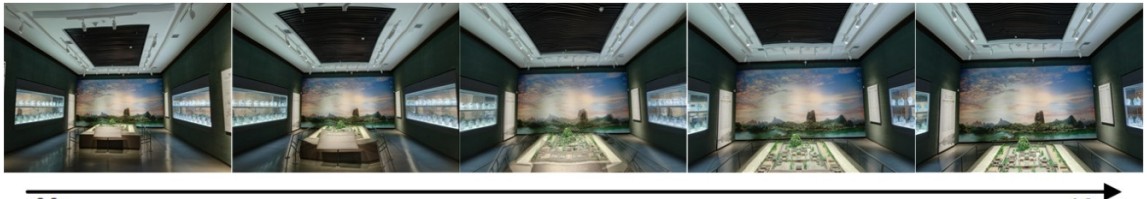

**Figure 7.** Visual performance of the 360° panoramic walkthrough system with smooth transitions between adjacent viewpoints. The panorama scene transitions from the current site to the next with the process ranging from 0.0 to 1.0.

## 3. Experiments and Results

### 3.1. Experimental Environment and Dataset

To comprehensively and effectively test the effect of this method, two datasets were collected by different acquisition equipment in different scenarios. Table 1 shows the collected panoramas and the corresponding acquisition equipment. The experimental data and results involved in this article can be obtained at https://github.com/zpc-whu/panoramic-transition.

**Table 1.** Experimental datasets.

| Dataset Name | Dataset-1. Traditional Panoramic Image Collection | Dataset-2. Mobile Mapping with Multiple Sensors Integrated | |
|---|---|---|---|
| Scenario | Digital Museum | Indoor Mapping | |
| acquisition equipment | the camera consists of 6 Nikon (AF-S 24–70 mm) lenses | the camera consists of 6 cheap cameras (XiaoYi lenses) | |
| resolution | 20,000 × 10,000 | 4096 × 2048 | |
| panoramic images | current station | 15 m | 10 m |
| | | 5 m | 3 m |
| | | 2 m | 1 m |
| | target station | | |

Dataset-1 consists of two panoramic pictures of a museum. These pictures are taken by a high-definition assembly camera with six lenses through the traditional station measurement in an digital museum construction project. The resolution can reach 20,000 × 10,000. The visual effect and accuracy testing of the proposed method is executed using these two panoramas.

Dataset-2 consists of seven panoramic pictures of a library, taken by a camera with six cheap XiaoYi lenses carried on the latest efficient indoor mobile measurement trolley during an indoor mapping project. The resolutions are 4096 × 2048. The robustness experiment of the proposed method is executed using these seven panoramas with six tests with distances from the cur-rent site to the destination site of 1 m, 2 m, 3 m, 5 m, 10 m, and 15 m.

### 3.2. Equations Visualization Performance

The visual effect and accuracy testing of the proposed method is executed using Dataset-1. The visualization performance is shown in Table 2. The first column in the table is the transition schedule. It proceeds from 0 to 1 to complete the entire transitional process. Particularly, views of 11 moments (1%, 10%, 20%, 30%, 40%, 50%, 60%, 70%, 80%,90%, and 99%) are selected to show the visual effects of the transitional methods. In contrast, the second column shows the results of the commonly used texture transparent gradient method, and the third column shows the results of the proposed method. The fourth column shows the deformation motion of the control triangular patches during the transition to better understand the principle of the proposed method.

**Table 2.** Visualization performance of the proposed method compared with the texture transparent gradient method.

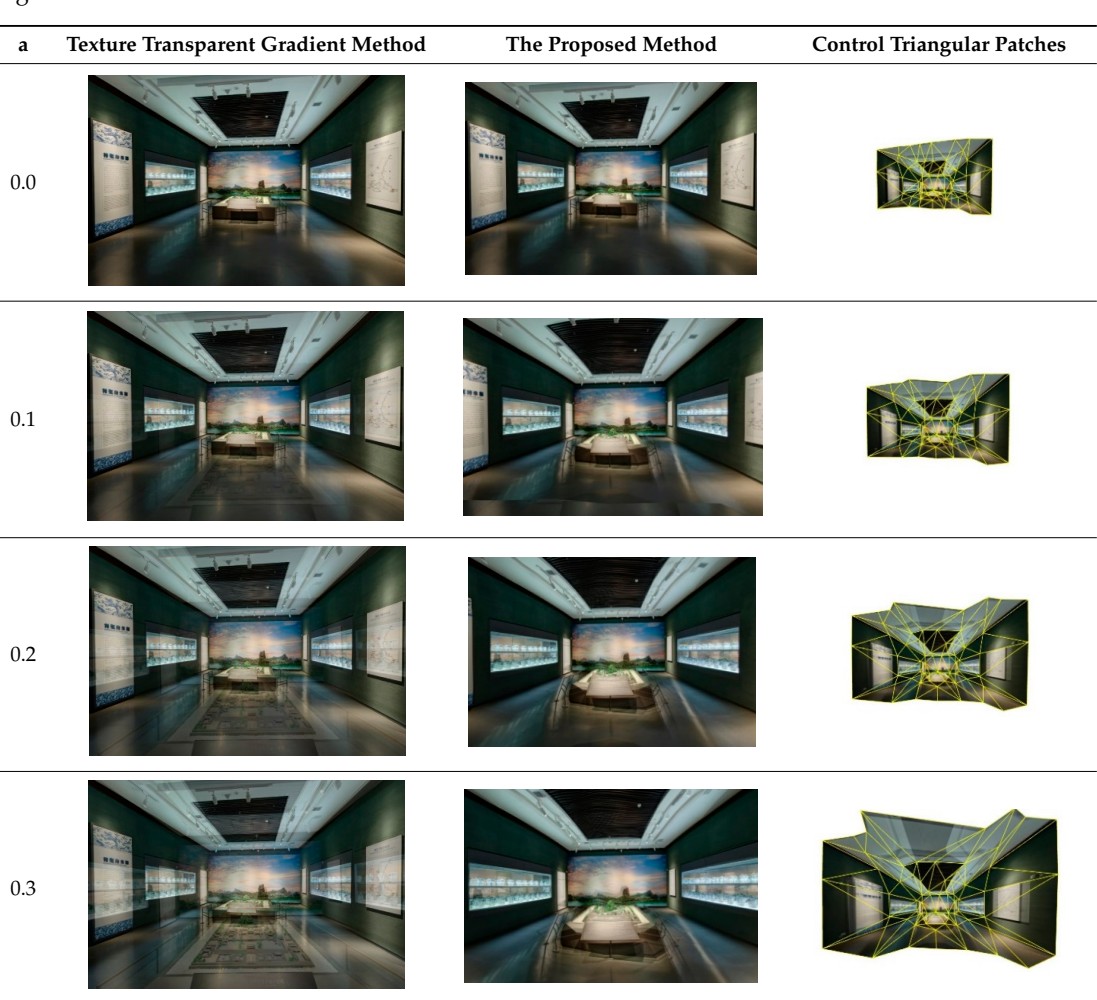

| a | Texture Transparent Gradient Method | The Proposed Method | Control Triangular Patches |
|---|---|---|---|
| 0.0 | | | |
| 0.1 | | | |
| 0.2 | | | |
| 0.3 | | | |

**Table 2.** *Cont.*

| a | Texture Transparent Gradient Method | The Proposed Method | Control Triangular Patches |
|---|---|---|---|
| 0.4 |  |  |  |
| 0.5 |  |  |  |
| 0.6 |  |  |  |
| 0.7 |  |  |  |
| 0.8 |  |  |  |
| 0.9 |  |  |  |
| 1.0 |  |  |  |

In Table 2, comparing the scenes in the third column with the scenes in the second column, the method proposed in this paper has obvious advantages over the existing the texture transparency gradient method. There was a slight ghosting only at the progress of approximately 50% using the proposed method, but there was a severe ghosting between 30% and 70% using the transparency gradient method. Moreover, observing the third columns in sequence, the user can experience a distinct visual movement (going in) without a perceptible visual distortion introduced by the existing model stretching method.

More specifically, the proposed transitional method can guarantee a smooth process on the cabinets on both sides, the front display screen and the simulated sand table on the floor, without ghosting and distortion. Therefore, the proposed method of panoramic transitions guided by the triangular patches is an effective seamless transitional solution for the indoor panoramic scene. Compared with the method of the transparency gradient and model stretching, this method can achieve extremely natural transitions of scenes in conditions with only panoramic data.

### 3.3. Applicability Performance

To determine the robustness, six experiments (jumping from different distances to the same site) were performed using Dataset-2 to verify the sensitivity of the proposed method to the distance between sites. The robustness is listed in Table 3. The first column is the distance from the start site to the target site of a transition, and six panoramic sites at different distances (1 m, 2 m, 3 m, 5 m, 10 m, and 15 m) from the target site are selected to show the robustness of the proposed transitional methods. The second column shows the control triangular patches extracted from each start site. The third column shows a snapshot of the user's view when transitioning to 50%. The fourth column shows the deformation introduced by the morphing mesh near the end of the transition. The first row shows the real target site for all six experiments.

**Table 3.** Robustness performance of the proposed method with six tests.

| Station Spacing | Transitional Execution to 1% | Transitional Execution to 50% | Transitional Execution to 99% |
|---|---|---|---|
| 0 m | — | — |  |
| 1 m |  |  |  |
| 2 m |  |  |  |

**Table 3.** *Cont.*

| Station Spacing | Transitional Execution to 1% | Transitional Execution to 50% | Transitional Execution to 99% |
|---|---|---|---|
| 3 m |  |  |  |
| 5 m |  |  |  |
| 10 m |  |  |  |
| 15 m |  |  |  |

In Table 3, from the second column, the smaller control area of the homonymic triangle patches with the increase in the distance between the two panorama sites is caused by the decreasing image overlap. More specifically, it is easy to extract the control triangle patches on the left side of the homonymic scene because of the commodious spatial distribution with objects farther away from the shooting site. However, it is challenging to extract the control triangle patches on the right side of the homonymic scene because of the large angular variations in the perspective caused by narrow spaces and occlusions of the objects in space. From the third column, the deformation of the generated panoramic scene increases, and the accuracy of the transitional model guided by the triangle patches decreases with the increase in the distance between the two panorama sites. In other words, the right side of the scene is significantly compressed, resulting in greater distortion.

In general, as the distance between the two sites increases from 1 m to 15 m, the overlap of the space scene decreases. This results in the control area that guides the transition decreasing, which increases the distortion of the generated scenes and worsens the transitional effect. However, the visual distortions resulting from the transitions within the 10 m range are acceptable in most application scenarios. The visual sense of movement is still maintained, and there is still an advantage over other transitional methods. Therefore, the proposed method can basically satisfy the smooth transition between two indoor panorama sites with a station spacing less than or equal to 10 m.

### 3.4. Efficiency Performance

To test the feasibility and efficiency of the proposed method, a variety of browser platforms extended to personal computers and mobile devices were used to run the single page application. Table 4 recodes the frame rate under different platforms and datasets when the scene is roaming and

transiting. The first column lists the four common browsers for testing, and the second and third columns are the frame rates in units of fps (frames per second).

**Table 4.** Efficiency performance using different data on various browsing platforms.

| Platform | Dataset-1 | | Dataset-2 | |
|---|---|---|---|---|
| | **Roam** | **Transition** | **Roam** | **Transition** |
| Windows 10 Chrome | 54 fps | 54 fps | 60 fps | 56 fps |
| Windows 10 Edge | 53 fps | 54 fps | 60 fps | 56 fps |
| XiaoMi Note 3 WeChat browser | 41 fps | 39 fps | 56 fps | 53 fps |
| Huawei Honour 5A QQ browser | 40 fps | 37 fps | 52 fps | 46 fps |

As Table 4 shows, different platforms keep different frame rates due to the different rendering capabilities, but there is slight difference in frame rates between roaming and transitioning on the same platform. This confirms that the proposed panorama transitional scheme does not slow down the rendering of panoramic scenes, and it is beyond the requirements of human visual observations (24 fps). Therefore, it is feasible and efficient.

## 4. Discussion

The seamless transition of the panoramic scene is achieved under the image morphing guidance of the triangular mesh generated by extracting the homonymous control points. From the extraction of homonymous feature points and the generation of triangular patches, to the calculation principles of triangular patch deformation and texture interpolation, to the design of the panoramic site transition algorithm and the implementation of the panoramic walkthrough system equipped with a dynamic morph layer, the proposed algorithm is described in detail. The further development of this methodology is the robust and smart matching of panoramic images.

The method is based on feature points, but it is difficult to find the feature points with practical significance in two panoramic images. Furthermore, the conventional line extraction and methods for eliminating mismatching points (such as distance anomaly and direction consistency) are also difficult to apply into panoramic images directly. Therefore, the development of feature extraction and matching between panoramic images is very important for the natural transition of panoramic scenes. The work of Carufel and Laganiere [22] and Kim and Park [23] can be a further reference regarding the matching of panoramic images.

As the transitional units, the control triangular patches play a vital role. For instance, the narrow triangle will cross the larger angle of view, resulting in larger distortions when we use the linear interpolation to simulate the continuous transition in the space. The object is broken when the same object is divided into different triangles, so it is necessary to make a reasonable division of the panoramic scene before doing a transition. In this regard, the latest research on restoring the depth and layout of houses from single panoramic images [24,25] is an inspiration to divide the panoramic space reasonably. Of course, the use of some auxiliary data can improve the transition, including planes in the scene extracted from the point cloud.

As the distance between stations increases, the shooting angle varies greatly. The control points will centrally appear in one cluster on the image or will be scattered over the upper and lower edges of the image. Such control triangular patches do not have strong reliability, frequently lead to greater deformation or confusion, and do not have good transitional visual effects. The analysis found that three major causes led to the above poor performance:

(1) the large distance between the sites with narrow space;
(2) objects are closer to the point of exposure and produce large occlusions;
(3) the depth distribution of the objects in space is diverse.

In view of the above problems, a shooting route with a relatively wide view should be a better choice.

Its natural smooth visual browsing experience is superior to the existing method of panoramic site transitions. Through the comprehensive test and analysis of its application scenarios and operational efficiency, the method is shown to be able to smoothly transition between adjacent panoramic sites collected by traditional station measurement or up-to-date mobile measurements. It also meets the needs of different applications, including panoramic space position navigation or high-resolution panoramic browsing. However, it should be noted that the method is not suitable for the scenes without obvious texture features or narrow and complicated scenes due to the invalid morph constraint caused by the rare feature points and small overlap.

## 5. Conclusions

In this paper, a dynamic panoramic image rendering algorithm for smooth transitions between adjacent viewpoints based on matched Delaunay triangular patches is proposed. The experiment shows that the proposed method seems to produce a visual sense of motion by the movement of vertexes and texture transparency changes in the morphing mesh. This work improves the user experience of the 360° panoramic roaming system and greatly improves the usability of indoor panoramic exhibition applications and indoor panoramic navigation applications. Furthermore, this technology can also be used in compressing video sequences with high compression rates, virtual tourism and entertainment, 3D scene displays of real estate and public security, etc.

**Author Contributions:** Conceptualization, Qingwu Hu and Mingyao Ai; methodology, Zhixiong Tang and Pengcheng Zhao; software, Zhixiong Tang and Pengcheng Zhao; validation, Zhixiong Tang and Pengcheng Zhao; formal analysis, Zhixiong Tang; investigation, Pengcheng Zhao; resources, Pengcheng Zhao; data curation, Pengcheng Zhao; writing—original draft preparation, Pengcheng Zhao; writing—review and editing, Qingwu Hu and Mingyao Ai; visualization, Zhixiong Tang; supervision, Qingwu Hu; Project administration, Qingwu Hu; funding acquisition, Mingyao Ai. All authors have read and agreed to the published version of the manuscript.

**Funding:** This work was partially supported by grants from the National Science Foundation of China (Grant no. 41701528), Science and Technology Planning Project of Guangdong, China (Grand No. 2017B020218001) and the Fundamental Research Funds for the Central Universities (Grant no.2042017kf0235).

**Conflicts of Interest:** The authors declare no conflict of interest.

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
