# Peer review of "A Smooth Transition Algorithm for Adjacent Panoramic Viewpoints Using Matched Delaunay Triangular Patches"

_ijgi, doi:10.3390/ijgi9100596_

Round 1

Reviewer 1 Report

This is my second viewing.  It seems that required changes have been made

Author Response

Dear Reviewer,

Thank you very much for your good suggestions on our manuscript. I will further improve the manuscript as soon as possible. Thank you again for your contribution to this manuscript.

Reviewer 2 Report

Overall this paper is quite interesting and the Smooth Transition algorithm is very good. Nevertheless, I believe that the following points will help the authors further improve that paper. This submission is a very interesting paper with a very applied topic and falls in the scope of the journal.  Although, it has some potential for improvement.

Introduction

  • The general description of the problem (Introduction) and the description of its importance for the science and the society could be further improved. I believe that the authors should establish why the topic of this work is quite important and needs further investigation.
  • The degree of innovativeness of the methodological approach is not convincingly demonstrated. Some more details about its innovative features could further improve the quality of this paper. For example, why is this paper likely to be cited in the future?
  • Indicating the current gap in knowledge and possible limitations, will further improve this paper.
  • A greater review of the current state of the art would further improve this work.

Method

  • A bit more text regarding the originality of this work and why it contains new results that significantly advance the research field.

Results

  • I believe that adding a bit more text on why the results of the method are satisfactory (evaluation approach) will increase the quality of this work
  • Could the results be more satisfactory if you have changed something in the methodology?
  • Are the results (or the method) sensitive to this specific study area?

Discussion

  • In the Discussion section I would have wished to see more information on the actual meaning of the findings and how the results add to the broader topic as well as the specific scientific field

Conclusion

  • The "Conclusions" section, could be further improved by describing the importance of this work, the highlight of potential further development of this methodology.

Author Response

Dear Reviewer,

Thank you for your Suggestions. According to your suggestion, we have made the following two improvements to the manuscript. First, in “Introduction” section, we added a paragraph that succinctly describes the Important significance of the topic of this work. Second, we further strengthen the description of the significance and expansion of the work in the discussion and conclusion

Reviewer 3 Report

Dear authors,

Thank you for addressing my comments and now the paper is possible for acceptance.

Best regards 

Author Response

Dear Reviewer,

Thank you very much for your good suggestions on our manuscript. I will further improve the manuscript as soon as possible. Thank you again for your contribution to this manuscript.

This manuscript is a resubmission of an earlier submission. The following is a list of the peer review reports and author responses from that submission.

Round 1

Reviewer 1 Report

This is a very well-written and illustrated article. I only have two areas for improvement:

1) I could not access your videos through the GitHub link.  The two files that are uploaded there are two small (111 KB) for a video.  There needs to be a easier to access video to actually see the results.  I would investigate what resources IJGI has available for video, or use Youtube.  A reader cannot properly judge the process without seeing a video or an actual panorama that you have made.

2)  In your conclusion, you state: 

The proposed method can produce a visual sense of motion by the movement of vertexes and texture transparency changes in the morphing mesh.

Let me ask on what basis you are making this determination?  Is it your own assessment?  In that case, modify the wording to say that it "seems to produce a visual sense of motion" but this impression would need to be verified through examination in a rigorous subject testing experimental procedure.

Overall, nice job and I look forward to seeing the video.

Author Response

To Reviewer 1:

1)  I could not access your videos through the GitHub link.  The two files that are uploaded there are two small (111 KB) for a video.  There needs to be a easier to access video to actually see the results.  I would investigate what resources IJGI has available for video, or use Youtube.  A reader cannot properly judge the process without seeing a video or an actual panorama that you have made.

Response: Thank you for your careful inspection. We upload the videos on Youtube. The video (https://youtu.be/R_lW1Xz8QNc) shows the problem of panoramic transition in a 360° panoramic walkthrough system. The video (https://youtu.be/xQyqiUuLPCI) shows our smooth panoramic transition using matched triangular patches.

2)  In your conclusion, you state: The proposed method can produce a visual sense of motion by the movement of vertexes and texture transparency changes in the morphing mesh. Let me ask on what basis you are making this determination?  Is it your own assessment?  In that case, modify the wording to say that it "seems to produce a visual sense of motion" but this impression would need to be verified through examination in a rigorous subject testing experimental procedure.

Response: Thank you for your advice. The determination was made by our subjective judgment. We have revised the wording according to your suggestion.

Reviewer 2 Report

This submission is a very interesting paper with a very applied topic and falls in the scope of the journal: International Journal of Geo-Information. Although, it has some potential for improvement.

Introduction

  • The general description of the problem (Introduction) and the description of its importance for the science and the society could be further improved.
  • The degree of innovativeness of the methodological approach is not convincingly demonstrated. Some more details about its innovative features could further improve the quality of this paper.
    • Why is this paper likely to be cited in the future?

Method

  • A bit more text regarding the originality of this work and why it contains new results that significantly advance the research field.
  • A bit more text regrading Synchronously Morph the Shape and Texture.

Results

  • I believe that adding a bit more text on why the results of the method are satisfactory (evaluation approach) will increase the quality of this work
  • Could the results be more satisfactory if you have changed something in the methodology?
  • Are the results (or the method) sensitive to this specific study area?

Discussion

  • In the Discussion section I would have wished to see more information on the actual meaning of the findings and how the results add to the broader topic as well as the specific scientific field

Conclusion

  • The "Conclusions" section, could be further improved by describing the importance of this work, the highlight of potential further development of this methodology.

Author Response

To Reviewer 2:

1)  Introduction: The general description of the problem (Introduction) and the description of its importance for the science and the society could be further improved. The degree of innovativeness of the methodological approach is not convincingly demonstrated. Some more details about its innovative features could further improve the quality of this paper. Why is this paper likely to be cited in the future?

Response: Thank you for your advice. The innovative feature of this paper is that we achieved the seamless transition of the panoramic scene by image morphing based triangular mesh, and the lines 78-100 describe the main principles of our method and its differences from other methods. The aim of this work is to improve the usability of indoor panoramic exhibition application and indoor panoramic navigation application.

2)  Method: A bit more text regarding the originality of this work and why it contains new results that significantly advance the research field. A bit more text regrading Synchronously Morph the Shape and Texture.

Response: Thank you for your advice. The shape morph of a triangle is implemented by linear interpolation in the Barycentric coordinates system. The texture interpolation of a triangle is implemented by superposition of two panoramic textures. The synchronously morph of the shape and the texture for a triangle described by Algorithm 1.

3)  Results: I believe that adding a bit more text on why the results of the method are satisfactory (evaluation approach) will increase the quality of this work. Could the results be more satisfactory if you have changed something in the methodology? Are the results (or the method) sensitive to this specific study area?

Response: Thank you for your advice. We use two different types of panoramic data to illustrate that our method is insensitive to relatively dense panoramic data. The results show that our method can satisfy the smooth transition between two indoor panorama sites with a station spacing less than or equal to 10 m. However, it should be noted that our method is not suitable for the scenes without obvious texture features or the narrow and complicated scenes, because of the invalid morph constraint caused by the rare feature points and small overlap. The relevant contents are described by section 3.3. Applicability Performance.

4)  Discussion: In the Discussion section I would have wished to see more information on the actual meaning of the findings and how the results add to the broader topic as well as the specific scientific field.

Response: Thank you for your advice. The 360° panoramic walkthrough system is widely used in 3D visualization, virtual reality and augmented reality because of its easy data acquisition and processing, low bandwidth requirements, high resolution, strong authenticity and good navigation. This work produces a visual sense of motion by the movement of vertexes and texture transparency changes in the morphing mesh. Its natural smooth visual browsing experience can improve the usability of panoramic exhibition application and panoramic navigation application.

5)  Conclusion: The "Conclusions" section, could be further improved by describing the importance of this work, the highlight of potential further development of this methodology.

Response: Thank you for your advice. We have enhanced the description in the conclusion section according to your suggestion. The importance of this work is that it improves the user experience of 360 ° panoramic roaming system and greatly improves the usability of indoor panoramic exhibition application and indoor panoramic navigation application. The potential further development of this methodology is the robust and smart matching of panoramic images, and lines 304-324 show the detailed description.

Reviewer 3 Report

Dear Authors,

Please find the attached file for my comments on your paper.

Best Regards 

Author Response

To Reviewer 3:

1)  Please add result details in the abstract section.

Response: Thank you for your advice. We have added a description of the results to the abstract according to your suggestion.

2)  The paper should add a related work section. Please consider some good journal papers and explain the existing technology.

Response: Thank you for your advice. We have not listed the Relevant Work sections separately. The related work has been described in the Introduction section. According to your opinion, we have added some high-quality references.

3)  Please add the rest of the paper is …. at the end of the Introduction section.

Response: Thank you for your advice. At the beginning of the Method section, we introduced the process steps of our method with illustrations in detail. So, we don't think it is necessary to add a description “the rest of the paper is ….” at the end of the Introduction section.

4)  Please check the paper format for mentioning the figures. I think Fig.--> Figure.

Response: Thank you for your careful inspection. We have changed all the words “Fig.” to “Figure.”

5)  Please check the Figures alignment. Please add a space before the figure placements.

Response: Thank you for your careful inspection. The manuscript used the word template (ijgi-template.dot) provided by IJGI. And after careful inspection, the format of first submission meets the requirements of the template.

6)  Please check the figures text font. It is not readable form.

Response: Thank you for your careful inspection. We re edited all the figures to make sure that the font of the figures is consistent with the font of the text.

7)  Please check the Figure 3 caption. Please update the alignment.

Response: Thank you for your careful inspection. The manuscript used the word template (ijgi-template.dot) provided by IJGI. And after careful inspection, the format of first submission meets the requirements of the template.

8)  Please check the equations 1 and 2 size. It is very small and not readable. Please update it.

Response: Thank you for your careful inspection. We have modified the size of the equations 1 and 2.

9)  Please check the typos carefully. I can see many mistakes from paper. Example: pre-forming→ preforming …

Response: Thank you for your careful inspection. We have corrected all the errors caused by newline hyphens in the manuscript.

10)  Please check the table 3 alignment.

Response: Thank you for your careful inspection. We have corrected all the errors caused by newline hyphens in the manuscript. The manuscript used the word template (ijgi-template.dot) provided by IJGI. And after careful inspection, the format of the table 3 meets the requirements of the template.

11)  The comparison result from the paper is not acceptable. The paper should compare the proposed method with conventional approaches and should validate their proposed method gives better performance than conventional approaches.

Response: Thank you for your advice. In fact, many 360° panoramic walkthrough systems have transition method of panoramic sites, but we find that there are few systems (codes or tools) and datasets that can be used free (open source or accessible interface) for comparison. Out of desperation, we can only choose the most basic method (the texture-transparent gradient method) to show the effect of our method.

12)  Please write the conclusion part in a single paragraph.

Response: Thank you for your careful inspection. Our Conclusion section is divided into two paragraphs. The first paragraph summarizes our method, and the second paragraph summarizes the experimental results of the method. We think the paragraph structure is appropriate.

13)  The paper looks like a thesis, not in a technical paper. Please refer good journal paper and update the format of the paper.

Response: Thank you for your advice. Because this research is inclined to engineering application, in order to write the whole workflow clearly, our article looks like a thesis. As for references, in the theoretical part, we refer to the relevant theoretical research in top journals of computer graphics such as TOG and CGF. In the practical application and system effect part, we can only find some articles in some application magazines. Anyway, we reviewed the relevant literature again and added several high-quality references.

14)  Please do a careful proof reading before next submission. The paper has so many mistakes and language errors.

Response: Thank you for your careful inspection. We have carefully checked and revised the manuscript.